# Anonymous Bandits for Multi-User Systems

**Hossein Esfandiari**
Google Research
esfandiari@google.com

**Vahab Mirrokni**
Google Research
mirrokni@google.com

**Jon Schneider**
Google Research
jschnei@google.com

## Abstract

In this work, we present and study a new framework for online learning in systems with multiple users that provide user anonymity. Specifically, we extend the notion of bandits to obey the standard $k$-anonymity constraint by requiring each observation to be an aggregation of rewards for at least $k$ users. This provides a simple yet effective framework where one can learn a clustering of users in an online fashion without observing any user's individual decision. We initiate the study of *anonymous bandits* and provide the first sublinear regret algorithms and lower bounds for this setting.

## 1 Introduction

In many modern systems, the system learns the behavior of the users by adaptively interacting with the users and adjusting to their responses. For example, in online advertisement, a website may show a certain type of ads to a user, observe whether the user clicks on the ads or not, and based on this feedback adjust the type of ads it serves the user. As part of this learning process, the system faces a dilemma between "exploring" new types of ads, which may or may not seem interesting to the user, and "exploiting" its knowledge of what types of ads historically seemed interesting to the user. This concept is very well studied in the context of online learning.

In principle, users benefit from a system that automatically adapts to their preferences. However, users may naturally worry about a system that observes all of their actions, and worry that the system may use this personal information against them or mistakenly reveal it to hackers or untrustworthy third parties. There therefore arises an additional dilemma between providing a better personalized user experience and acquiring the users' trust.

We study the problem of online learning in multi-user systems under a version of anonymity inspired by *k-anonymity* [Sweeney, 2002]. In its most general form, $k$-anonymity is a property of anonymous data guaranteeing that for every data point, there exist at least $k - 1$ other indistinguishable data points in the dataset. Although there exist more recent notions of privacy and anonymity with stronger guarantees (e.g. various forms of differential privacy), $k$-anonymity is a simple and practical notion of anonymity that remains commonly employed in practice and enforced in various legal settings [Goldsteen et al., 2021, Saf, November 2019, Slijepčević et al., 2021].

To explain our notion of anonymity, consider the online advertisement application mentioned earlier. In online advertisement, when a user visits a website the website selects a type of ad and shows that user an ad of that type. The user may then choose to click on that ad or not, and a reward is paid out based on whether the user clicks on the ad or not (this reward may represent either the utility of the user or the revenue of the online advertisement system). Note that the ad here is chosen by the website, and the fact that the website assigns a user a specific type of ad is not something we intend to hide from the website. However, the decision to click on the ad is made by the user. We intend to protect these individual decisions, while allowing the website to learn what general types of ads each user is interested in. In particular, we enforce a form of group level $C$-anonymity on these decisions, by forcing the system to group users into groups of at least $C$ users and to treat each group equally by

assigning all users in the same group the same ad and only observing the total aggregate reward (e.g. the total number of clicks) of these users.

More formally, we study this version of anonymity in a simple model of online learning based on the popular multi-armed bandit setting. In the classic (stochastic) multi-armed bandits problem, there is a learner with some number $K$ of potential actions ("arms"), where each arm is associated with an unknown distribution of rewards. In each round (for $T$ rounds) the learner selects an arm and collects a reward drawn from a distribution corresponding to that arm. The goal of the learner is to maximize their total expected reward (or equivalently, minimize some notion of regret).

In our multi-user model, a centralized learner assigns $N$ users to $K$ arms, and each rewards for each user/arm pair are drawn from some fixed, unknown distribution. Each round the learner proposes an assignment of users to arms, upon which each user receives a reward from the appropriate user/arm distribution. However, the learner is only allowed to record feedback about these rewards (and use this feedback for learning) if they perform this assignment in a manner compatible with $C$-*anonymity*. This entails partitioning the users into groups of size at least $C$, assigning all users in each group to the same arm, and only observing this group's aggregate rewards for this arm. For example, if $C = 3$ in one round we may combine users 1, 2 and 4 into a group and assign them all to arm 3; we would then observe as feedback the aggregate reward $r_{13} + r_{23} + r_{43}$, where $r_{ij}$ represents the reward that user $i$ experienced from arm $j$ this round. The goal of the learner is to maximize the total reward by efficiently learning the optimal action for each user, while at the same time preserving the anonymity of individual rewards users experienced in specific rounds. See Section 2 for a more detailed formalization of the model.

## 1.1 Our results

In this paper we provide low-regret algorithms for the anonymous bandit setting described above. We present two algorithms which operate in different regimes (based on how users cluster into their favorite arms):

- If for each arm $j$ there are at least $U \geq C + 1$ users for which arm $j$ is that user's optimal arm, then Algorithm 1 incurs regret at most $\tilde{O}(NC\sqrt{\alpha KT})$, where $\alpha = \max(1, \lceil K(C+1)/U \rceil)$.
- If there is no such guarantee (but $N > C$), then Algorithm 2 incurs regret at most $\tilde{O}(C^{2/3}K^{1/3}T^{2/3})$.

We additionally prove the following corresponding lower bounds:

- We show (Theorem 3) that the regret bound of Algorithm 1 is tight for any algorithm to within a factor of $K\sqrt{C}$ (and to within a factor of $\sqrt{C}$ for $U \geq K(C + 1)$).
- We show (Theorem 4) that the dependence on $T^{2/3}$ in Algorithm 2 is necessary in the absence of a lower bound on $U$.

The main technical contribution of this work is the development/analysis of Algorithm 1. The core idea behind Algorithm 1 is to use recent algorithms developed for the problem of *batched bandits* (where instead of $T$ rounds of adaptivity, users are only allowed $B \ll T$ rounds of adaptivity) to reduce this learning problem to a problem in combinatorial optimization related to decomposing a bipartite weighted graph into a collection of degree-constrained bipartite graphs. We then use techniques from combinatorial optimization and convex geometry to come up with efficient approximation algorithms for this combinatorial problem.

## 1.2 Related work

The bandits problem has been studied for almost a century [Thompson, 1933], and it has been extensively studied in the standard single user version [Audibert et al., 2009a,b, Audibert and Bubeck, 2010, Auer et al., 2002, Auer and Ortner, 2010, Bubeck et al., 2013, Garivier and Cappé, 2011, Lai and Robbins, 1985]. There exists recent work on bandits problem for systems with multiple users [Bande and Veeravalli, 2019a,b, Bande et al., 2021, Vial et al., 2021, Buccapatnam et al., 2015, Chakraborty et al., 2017, Kolla et al., 2018, Landgren et al., 2016, Sankararaman et al., 2019]. These papers study this problem from game theoretic and optimization perspectives (e.g. studying coordination /

competition between users) and do not consider anonymity. To the best of our knowledge this paper is the first attempt to formalize and study multi-armed bandits with multiple users from an anonymity perspective.

Learning how to assign many users to a (relatively) small set of arms can also be thought of as a clustering problem. Clustering of users in multi-user multi-arm bandits has been previously studied. Maillard and Mannor [2014] first studied sequentially clustering users, which was later followed up by other researchers [Nguyen and Lauw, 2014, Gentile et al., 2017, Korda et al., 2016]. Although these works, similar to us, attempt to cluster the users, they are allowed to observe each individual's reward and optimize based on that, which contradicts our anonymity requirement.

One technically related line of work that we heavily rely on is recent work on batched bandits. In the problem of batched bandits, the learner is prevented from iteratively and adaptively making decisions each round; instead the learning algorithm runs in a small number of "batches", and in each batch the learner chooses a set of arms to pull, and observes the outcome at the end of the batch. Batched multi-armed bandits were initially studied by Perchet et al. [2016] for the particular case of two arms. Later Gao et al. [2019] studied the problem for multiple arms. Esfandiari et al. [2019] improved the result of Gao et al. and extended it to linear bandits and adversarial multi-armed bandits. Later this problem was studied for batched Thompson sampling [Kalkanli and Ozgur, 2021, Karbasi et al., 2021], Gaussian process bandit optimization [Li and Scarlett, 2021] and contextual bandits [Zhang et al., 2021, Zanette et al., 2021].

In another related line of work, there have been several successful attempts to apply different notions of privacy such as differential privacy to multi-armed bandit settings [Tossou and Dimitrakakis, 2016, Shariff and Sheffet, 2018, Dubey and Pentland, 2020, Basu et al., 2019]. While these papers provide very promising guarantees of privacy measures, they focus on single-user settings. In this work we take advantage of the fact that there are several similar users in the system, and use this to provide guarantees of anonymity. Anonymity and privacy go hand in hand, and in a practical scenario, both lines of works can be combined to provide a higher level of privacy.

Finally, our setting has some similarities to the settings of stochastic linear bandits [Dani et al., 2008, Rusmevichientong and Tsitsiklis, 2010, Abbasi-Yadkori et al., 2011] and stochastic combinatorial bandits [Chen et al., 2013, Kveton et al., 2015]. For example, the superficially similar problem of assigning $N$ users to $K$ arms each round so that each arm has at least $C$ users assigned to it (and where you get to observe the total reward per round) can be solved directly by algorithms for these frameworks. However, although such assignments are $C$-anonymous, there are important subtleties that prevent us from directly applying these techniques in our model. First of all, we do not actually constrain the assignment of users to arms – rather, our notion of anonymity constrains what feedback we can obtain from such an assignment (e.g., it is completely fine for us to assign zero users to an arm, whereas in the above model no actions are possible when $N < CK$). Secondly, we obtain more nuanced feedback than is assumed in these frameworks (specifically, we get to learn the reward of each group of $\geq C$ users, instead of just the total aggregate reward). Nonetheless it is an interesting open question if any of these techniques can be applied to improve our existing regret bounds (perhaps some form of linear bandits over the anonymity polytopes defined in Section 3.4.2).

## 2  Model and preliminaries

**Notation.**  We write $[N]$ as shorthand for the set $\{1, 2, \ldots, N\}$. We write $\tilde{O}(\cdot)$ to suppress any poly-logarithmic factors (in $N$, $K$, $C$, or $T$) that arise. We say a random variable $X$ is $\sigma^2$-subgaussian if the mean-normalized variable $Y = X - \mathbb{E}[x]$ satisfies $\mathbb{E}[\exp(sY)] \leq \exp(\sigma^2 s^2/2)$ for all $s \in \mathbb{R}$.

Proofs of most theorems have been postponed to Appendix C of the Supplemental Material in interest of brevity.

### 2.1  Anonymous bandits

In the problem of *anonymous bandits*, there are $N$ users. Each round (for $T$ rounds), our algorithm must assign each user to one of $K$ arms (multiple users can be assigned to the same arm). If user $i$ plays arm $j$, they receive a reward drawn independently from a 1-subgaussian distribution $\mathcal{D}_{i,j}$ with (unknown) mean $\mu_{i,j} \in [0, 1]$. We would like to minimize their overall expected regret of our algorithm. That is, if user $i$ receives reward $r_{i,t}$ in round $t$, we would like to minimize

$$\mathbf{Reg} = T \sum_{i=1}^{N} \max_j \mu_{i,j} - \mathbb{E}\left[\sum_{t=1}^{T}\sum_{i=1}^{N} r_{i,t}\right].$$

Thus far, this simply describes $n$ independent parallel instances of the classic multi-armed bandit problem. We depart from this by imposing an *anonymity constraint* on how the learner is allowed to observe the users' feedback. This constraint is parameterized by a positive integer $C$ (the minimum group size). In a given round, the learner may partition a subset of the users into groups of size at least $C$, under the constraint that the users within a single group must all be playing the same arm during this round. For each group $G$, the learner then receives as feedback the total reward $\sum_{i \in G} r_{i,t}$ received by users in this group. Note that not all users must belong to a group (the learner simply receives no feedback on such users), and the partition into groups is allowed to change from round to round.

Without any constraint on the problem instance, it may be impossible to achieve sublinear regret (see Section 3.6). We therefore additionally impose the following *user-cluster assumption* on the users: each arm $j$ is the optimal arm for at least $U$ users. Such an assumption generally holds in practice (e.g., in the regime where there are many users but only a few classes of arms). This also prevents situations where, e.g., only a single user likes a given arm but it is hard to learn this without allocating at least $C$ users to this arm and sustaining significant regret. Typically we will take $U > C$; when $U \leq C$ the asymptotic regret bounds for our algorithms may be worse (see Section 3.5).

### 2.2 Batched stochastic bandits

Our main tool will be algorithms for *batched stochastic bandits*, as described in Gao et al. [2019]. For our purposes, a batched bandit algorithm is an algorithm for the classical multi-armed bandit problem that proceeds in $B$ stages ("batches") where the $b$th stage has a predefined length of $D_b$ rounds (with $\sum_b D_b = T$). At the beginning of each stage $b$, the algorithm outputs a non-empty subset of arms $A_b$ (representing the set of arms the algorithm believes might still be optimal). At the end of each stage, the algorithm expects at least $D_b/|A_b|$ independent instances of feedback from arm $j$ for each $j \in S$; upon receiving such feedback, the algorithm outputs the subset of arms $A_{b+1}$ to explore in the next batch.

In Gao et al. [2019], the authors design a batched bandit algorithm they call BaSE (batched successive-elimination policy); for completeness, we reproduce a description of their algorithm in Appendix A. When $B = \log\log T$, their algorithm incurs a worst-case expected regret of at most $\tilde{O}(\sqrt{KT})$. In our analysis, we will need the following slightly stronger bound on the behavior of BaSE:

**Lemma 1.** *Set $B = \log\log T$. Let $\mu^* = \max_{j \in [K]} \mu_j$, and for each $j \in [K]$ let $\Delta_j = \mu^* - \mu_j$. Then for each $1 \leq b \leq B$, we have that:*

$$D_b \cdot \mathbb{E}\left[\max_{j \in A_b} \Delta_j\right] = \tilde{O}(\sqrt{KT}).$$

In other words, Lemma 1 bounds the expected regret in each batch, even under the assumption that we receive the reward of the worst active arm each round (even if we ask for feedback on a different arm).

It will also be essential in the analysis that follows that the total number of rounds $D_b$ in the $b$th batch only depends on $b$ and is independent of the feedback received thus far (in the language of Gao et al. [2019], the grid used by the batched bandit algorithm is *static*). This fact will let us run several instances of this batched bandit algorithm in parallel, and guarantees that batches for different instances will always have the same size.

## 3 Anonymous Bandits

### 3.1 Feedback-eliciting sub-algorithm

Our algorithms for anonymous bandits will depend crucially on the following sub-algorithm, which allows us to take a matching from users to arms and recover (in $O(C)$ rounds) an unbiased estimate

of each user's reward (as long as that user is matched to a popular enough arm). More formally, let $\pi : [N] \to [K]$ be a matching from users to arms. We will show how to (in $2C + 2$ rounds) recover an unbiased estimate of $\mu_{i,\pi(i)}$ for each $i$ such that $|\pi^{-1}(\pi(i))| \geq C + 1$ (i.e. $i$ is matched to an arm that at least $C$ other users are matched to).

The main idea behind this sub-algorithm is simple; for each user, we will get a sample of the total reward of a group containing the user, and a sample of the total reward of the same group but minus this user. The difference between these two samples is an unbiased estimate of the user's reward. More concretely, we follow these steps:

1. Each round (for $2C + 2$ rounds) the learner will assign user $i$ to arm $\pi(i)$. However, the partition of users into groups will change over the course of these $2C + 2$ rounds.

2. For each arm $j$ such that $|\pi^{-1}(j)| \geq C + 1$, partition the users in $\pi^{-1}(j)$ into groups of size at least $C + 1$ and of size at most $2C + 1$. Let $G_1, G_2, \ldots, G_S$ be the set of groups formed in this way (over all arms $j$). In each group, order the users arbitrarily.

3. In the first round, the learner reports the partition into groups $\{G_1, G_2, \ldots, G_S\}$. For each group $G_s$, let $r_{s,0}$ be the total aggregate reward for group $G_s$ this round.

4. In the $k$th of the next $2C + 1$ rounds, the learner reports the partition into groups $\{G_{1,k}, G_{2,k}, \ldots, G_{S,k}\}$, where $G_{s,k}$ is formed from $G_s$ by removing the $k$th element (if $k > |G_s|$, then we set $G_{s,k} = G_s$). Let $r_{s,k}$ be the total aggregate reward from $G_{s,k}$ reported this round.

5. If user $i$ is the $k$th user in $G_s$, we return the estimate $\hat{\mu}_{i,\pi(i)} = r_{s,0} - r_{s,k}$ of the average reward for user $i$ and arm $\pi(i)$.

**Lemma 2.** *In the above procedure, $\mathbb{E}[\hat{\mu}_{i,\pi(i)}] = \mu_{i,\pi(i)}$ and $\hat{\mu}_{i,\pi(i)}$ is an $O(C)$-subgaussian random variable.*

### 3.2 Anonymous decompositions of bipartite graphs

The second ingredient we will need in our algorithm is the notion of an anonymous decomposition of a weighted bipartite graph. Intuitively, by running our batched stochastic bandits algorithm, at the beginning of each batch we will obtain a demand vector for each user (representing the number of times that user would like feedback on each of the $m$ arms). Based on this, we want to generate a collection of assignments (from users to arms) which guarantee that we obtain (while maintaining our anonymity guarantees) the requested amount of information for each user/arm pair.

Formally, we represent a weighted bipartite graph as a matrix of $nm$ non-negative entries $w_{i,j}$ (representing the number of instances of feedback user $i$ desires from arm $j$). We will assume that for each $i$, $\sum_j w_{i,j} > 0$ (each user is interested in at least one arm). A *C-anonymous decomposition* of this graph is a collection of $R$ assignments $M_1, M_2, \ldots, M_R$ from users to arms (i.e., functions from $[N]$ to $[K]$) that satisfies the following properties:

1. A user is never assigned to an arm for which they have zero demand. That is, if $M_r(i) = j$, then $w_{i,j} \neq 0$.

2. If $M_r(i) = j$, and $|M_r^{-1}(j)| \geq C + 1$, we say that matching $M_r$ is *informative* for the user/arm pair $(i, j)$. (Note that this is exactly the condition required for the feedback-eliciting sub-algorithm to output the unbiased estimate $\hat{\mu_{i,j}}$ when run on assignment $M_r$.) For each user/arm pair $(i, j)$, there must be at least $w_{i,j}$ informative assignments.

The weighted bipartite graphs that concern us come from the parallel output of $N$ batched bandit algorithms (described in Section 2.2) and have additional structure. These graphs can be described by a positive total demand $D$ and a non-empty demand set $A_i \subseteq [K]$ for each user $i$ (describing the arms of interest to user $i$). If $j \in A_i$, then $w_{i,j} = D/|A_i|$; otherwise, $w_{i,j} = 0$. To distinguish graphs with the above structure from generic weighted bipartite graphs, we call such graphs *batched graphs*.

Moreover, for each user $i$, let $j^*(i)$ be the optimal arm for user $i$ (i.e., $j^*(i) = \arg\max_i \mu_i$). In the algorithm we describe in the next section, with high probability, $j^*(i)$ will always belong to $A_i$. Moreover, a user-cluster assumption of $U$ implies that, for any arm $j$, $|j^{*-1}(j)| \geq U$. This means that in batched graphs that arise in our algorithm, there will exist an assignment where each

user $i$ is assigned to an arm in $A_i$, and each arm $j$ has at least $U$ assigned users. We therefore call graphs that satisfy this additional assumption $U$-*batched graphs*. Note that this assumption also allows us to lower bound the degrees of arms in this bipartite graph. Specifically, for each arm $j$, let $B_j = \{i \in [N] \mid j \in A_i\}$. Then a user-cluster assumption of $U$ directly implies that $|B_j| \geq U$.

In general, our goal is to minimize the number of assignments $R$ required in such a decomposition (since each assignment corresponds to some number of rounds required). We call an algorithm that takes in a $U$-batched graph and outputs a $C$-anonymous decomposition of that graph an *anonymous decomposition algorithm*, and say that it has approximation ratio $\alpha(C, U) \geq 1$ if it generates an assignment with at most $\alpha(C, U) \cdot D + O(NK)$ total assignments. This additive $O(NK)$ is necessary for technical reasons, but in our algorithm, $D$ will always be much larger than $K$ (we will have $D \geq \sqrt{T}$), so this can be thought of as an additive $o(D)$ term.

Later, in Section 3.4, we explicitly describe several anonymous decomposition algorithms and their approximation guarantees. In the interest of presenting the algorithm, we will assume for now we have access to a generic anonymous decomposition algorithm Decompose with approximation ratio $\alpha(C, U)$.

### 3.3 An algorithm for anonymous bandits

We are now ready to present our algorithm for anonymous bandits. The main idea behind this algorithm (detailed in Algorithm 1) is as follows. Each of the $N$ users will run their own independent instance of BaSE with $B = \log \log T$ synchronized batches. During batch $b$, BaSE requires each user $i$ to get a total of $D_b$ instances of feedback on a set $A_{i,b}$ of arms which are alive for them. These sets $A_{i,b}$ (with high probability) define a $U$-batched graph, so we can use an anonymous decomposition algorithm to construct a $C$-anonymous decomposition of this graph into at most $\alpha(C, U)D_b$ assignments. We then run the feedback-eliciting sub-algorithm on each assignment, getting one unbiased estimate of the reward for each user/arm pair for which the assignment is informative.

The guarantees of the $C$-anonymous decomposition mean that this process gives us $D_b$ total pieces of feedback for each user, evenly split amongst the arms in $A_{i,b}$. We can therefore pass this feedback along to BaSE, which will eliminate some arms and return the set of alive arms for user $i$ in the next batch.

---

**Algorithm 1:** Low-regret algorithm for anonymous bandits.

---

**Input:** Anonymity parameter $C$, a lower bound on the user-cluster assumption $U$, time-horizon $T$, number of users $N$, and number of arms $K$. We additionally assume we have access to an anonymous decomposition algorithm Decompose with approximation factor $\alpha = \alpha(C, U)$.

For each user $i \in [N]$, initialize an instance of BaSE with a time horizon of $T' = \frac{T}{\alpha(2C+2)}$ and $B = \log \log T$ batches. Let $D_1, D_2, \ldots, D_b$ be the corresponding batch sizes (with $\sum_b D_b = T'$). Let $A_{i,b}$ be the set of active arms for user $i$ during batch $b$.

**for** $b \leftarrow 1$ **to** $B$ **do**

    If $D_b$ and $(A_{1,b}, A_{2,b}, \ldots, A_{n,b})$ do not define a $U$-batched graph, abort the algorithm (this means we have eliminated the optimal arm for a user, which can only happen with negligible probability).

    Run Decompose on $D_b$ and $(A_{1,b}, A_{2,b}, \ldots, A_{n,b})$ to get a $C$-anonymous decomposition into at most $R_b = \alpha D_b$ assignments. Let $M_r$ be the $r$th such assignment.

    **for** $r \leftarrow 1$ **to** $R_b$ **do**

        | Over $(2C + 2)$ rounds, run the feedback-soliciting sub-algorithm (Section 3.1) on $M_r$ to get estimates $\hat{\mu}_{i, M_r(i)}$ for all users $i$ for which $M_r$ is informative.

    **end**

    For each user $i$, we are guaranteed to receive (by the guarantees of Decompose) at least $\frac{D_b}{|A_{i,b}|}$ independent samples of feedback $\hat{\mu}_{i,j}$ for each arm $j \in A_{i,b}$. Pass these samples to user $i$'s instance of BaSE and receive $A_{i,b+1}$ in response (unless this is the last batch).

**end**

---

**Theorem 1.** *Algorithm 1 incurs an expected regret of at most $\tilde{O}(NC\sqrt{\alpha KT})$ for the anonymous bandits problem.*

One quick note on computational complexity: note that we only run Decompose once every *batch*; in particular, at most $\log \log T$ times. This allows us to efficiently implement Algorithm 1 even for complex choices of Decompose that may require solving several linear programs.

### 3.4 Algorithms for constructing anonymous decompositions

#### 3.4.1 A greedy method

We begin with perhaps the simplest method for constructing an anonymous decomposition, which achieves an approximation ratio $\alpha(C, U) = K$ as long as $U \geq C + 1$. To do this, for each arm $j \in [K]$, consider the assignment $M_j$ where all users $i$ with $j \in A_i$ (i.e., users with any interest in arm $j$) are matched to arm $j$, and other users are arbitrarily assigned to arms in their active arm set. Our final decomposition contains $D$ copies of $M_j$ for each $j \in [K]$ (for a total of $KD$ assignments).

Note that since $U \geq C + 1$, there will be at least $C + 1$ users matched to arm $j$ in $M_j$, and therefore $M_j$ will be informative for all users $i$ with $j \in A_i$. Since we repeat each assignment $D$ times, we will have at least $D$ informative assignments for every valid user/arm pair, and therefore this is a valid $C$-anonymous decomposition for the original batched graph.

Substituting this guarantee into Theorem 1 gives us an anonymous bandit algorithm with expected regret $\tilde{O}(NCK\sqrt{T})$.

#### 3.4.2 The anonymity polytope

As $U$ grows larger than $C$, it is possible to attain even better approximation guarantees. In this section we will give an anonymous decomposition algorithm that applies techniques from combinatorial optimization to attain the following guarantees:

- If $U \geq K(C + 1)$, then $\alpha(C, U) = 1$.
- If $(C + 1) \leq U \leq K(C + 1)$, then $\alpha(C, U) = \left\lceil \frac{K(C+1)}{U} \right\rceil$.

To gain some intuition for how this is possible, assume $U = K(C + 1)$, and consider the randomized algorithm which matches each user $i$ to a random arm in $A_i$ each turn. In expectation, after $D$ rounds of this, user $i$ will be matched to each arm $j \in A_i$ exactly $D/|A_i|$ times (as user $i$ desires). Moreover, since $U \geq K(C + 1)$, each arm $j$ has at least $K(C + 1)$ candidate users that can match to it. Each of these users matches to arm $j$ with probability at least $1/K$, so in expectation at least $C + 1$ users match to arm $j$, and therefore the feedback from arm $j$ is informative "in expectation".

The catch with this method is that it is possible (and even reasonably likely) for fewer than $C + 1$ users to match to a given arm $j$, and in this case we receive no feedback for user $i$. While it is possible to adapt this method to work with high probability, this requires additional logarithmic factors in either $\alpha$ or the user-cluster bound $U$, and even then has some probability of failure. Instead, we present a deterministic algorithm which can exactly achieve the guarantees above by geometrically "rounding" the above randomized matching into a small weighted collection of deterministic matchings.

We define the *C-anonymity polytope* $\mathcal{P}_C \subseteq \mathbb{R}^{N \times K}$ to be the convex hull of all binary vectors $v \in \{0, 1\}^{N \times K}$ that satisfy the following conditions:

- For each $i$, $\sum_j v_{ij} \in \{0, 1\}$.
- For each $j$, either $\sum_i v_{ij} \geq C + 1$ or $\sum_i v_{ij} = 0$.

We can interpret each such vertex $v$ as a single assignment in a $C$-anonymous decomposition, where $v_{ij} = 1$ iff we get feedback on the user/arm pair $(i, j)$ (so we must match user $i$ to $j$, and at least $C + 1$ users must be matched to arm $j$).

Now, for a fixed $U$-batched graph $G$, let $w \in [0, 1]^{N \times K}$ be the weights of this $G$ normalized by the demand $D$: so $w_{ij} = 1/|A_i|$ if $j \in A_i$, and $w_{ij} = 0$ otherwise. It turns out that we can reduce (via Caratheodory's theorem) the problem of finding a $C$-anonymous decomposition of $G$ into finding the maximal $\beta$ for which $\beta w \in \mathcal{P}_C$.

**Lemma 3.** *If for some $\beta > 0$, $\beta w \in \mathcal{P}_C$, then there exists a $C$-anonymous decomposition of $G$ into at most $\frac{1}{\beta}D + NK + 1$ assignments. Similarly, if there exists a $C$-anonymous decomposition of $G$ into $\alpha D$ assignments, then $\frac{1}{\alpha}w \in \mathcal{P}_C$.*

In a sense, Lemma 3 provides an "optimal" algorithm for the problem of finding $C$-anonymous decompositions. There are two issues with using this algorithm in practice. The first – an interesting open question – is that we do not understand the approximation guarantees of this decomposition algorithm (although they are guaranteed to be at least as good as every algorithm we present here).

**Open Problem 1.** *For a $U$-batched graph $G$, let $\beta(G)$ be the maximum value of $\beta$ such that $\beta w \in \mathcal{P}_C$ (where $w$ is the weight vector associated with $C$). What is $\max \beta(G)$ over all $U$-batched graphs? Is it $\Omega(1)$?*

The second is that, computationally, it is not clear if there is an efficient way to check whether a point belongs to $\mathcal{P}_C$ (let alone write it as a convex combination of the vertices of $\mathcal{P}_C$). We will now decompose $\mathcal{P}_C$ into the convex hull of a collection of more tractable polytopes; while it will still be hard to e.g. check membership in $\mathcal{P}_C$, this will help us efficiently compute decompositions that provide the guarantees at the beginning of this section.

For a subset $S \subseteq [K]$ of arms, let $\mathcal{P}_C(S) \subseteq \mathbb{R}^{N \times K}$ be the convex hull of the binary vectors that satisfy the following conditions:

- For each $i$, $\sum_j v_{ij} \in \{0, 1\}$.
- If $j \in S$, then $\sum_i v_{ij} \geq C + 1$.
- If $j \notin S$, then $\sum_i v_{ij} = 0$.

By construction, each vertex of $\mathcal{P}_C$ appears as a vertex of some $\mathcal{P}_C(S)$ and each vertex of $\mathcal{P}_C(S)$ belongs to $\mathcal{P}_C$, so $\mathcal{P}_C = \text{conv}(\{\mathcal{P}_C(S) \mid S \subseteq [K]\})$. We now claim that we can write each polytope $\mathcal{P}_C(S)$ as the intersection of a small number of halfspaces (and therefore check membership efficiently). In particular, we claim that $x \in \mathbb{R}^{N \times K}$ belongs to $\mathcal{P}_C(S)$ iff it satisfies the following linear constraints:

$$
\begin{aligned}
0 \leq\ & x_{ij} \leq 1, && \forall\, i \in [N], j \in [K] \\
& \sum_{j \in [K]} x_{ij} \leq 1, && \forall\, i \in [N] \\
& \sum_{i \in [N]} x_{ij} \geq C + 1, && \forall\, j \in S \\
& \sum_{i \in [N]} x_{ij} = 0, && \forall\, j \in [K] \setminus S.
\end{aligned}
\tag{1}
$$

**Lemma 4.** *A point $x \in \mathbb{R}^{N \times K}$ belongs to $\mathcal{P}_C(S)$ iff it satisfies the constraints in* (1).

We can now prove the guarantees at the beginning of the section. We start with the case where $U \geq K(C + 1)$. Here we show (via similar logic to the initial randomized argument) that $w \in \mathcal{P}_C([K])$.

**Lemma 5.** *If $U \geq K(C + 1)$, then $w \in \mathcal{P}_C([K])$.*

Applying Caratheodory's theorem, we immediately obtain a $C$-anonymous decomposition from Lemma 5

**Corollary 1.** *If $U \geq K(C+1)$, there exists a $C$-anonymous decomposition into at most $D+KC+1$ assignments. Moreover, it is possible to find this decomposition efficiently.*

When $C + 1 \leq U \leq K(C + 1)$, we first arbitrarily partition our arms into $\alpha = \lceil K(C + 1)/U \rceil$ blocks $S_1, S_2, \ldots, S_\alpha$ of at most $U/(C + 1)$ vertices each. We then show how to write $w$ as a linear combination of $\alpha$ points, one in each of the polytopes $\mathcal{P}_C(S_a)$.

**Lemma 6.** *There exist points $w^{(a)} \in \mathcal{P}_C(S_a)$ for $1 \leq a \leq \alpha$ such that $w \leq \sum_{i=1}^{\alpha} w^{(a)}$.*

Likewise, we can again apply Caratheodory's theorem to obtain a $C$-anonymous decomposition from Lemma 6.

**Corollary 2.** *If $(C + 1) \leq U \leq K(C + 1)$, there exists a $C$-anonymous decomposition into at most $\alpha D + NK + 1$ assignments, where $\alpha = \lceil K(C + 1)/U \rceil$. Moreover, it is possible to find this decomposition efficiently.*

### 3.5   Settings without user-clustering

The previous algorithms we have presented for anonymous bandits rely heavily on the existence of a user-cluster assumption $U$. In this section we present an algorithm (Algorithm 2) for anonymous bandits which works in the absence of any user-cluster assumption as long as $N > C$. This comes at the cost of a slightly higher regret bound which scales as $T^{2/3}$ – as we shall see in Section 3.6, this dependence is in some sense necessary.

---

**Algorithm 2:** Explore-then-commit algorithm for anonymous bandits without a user-clustering assumption.

---

**Input:** Anonymity parameter $C$, time-horizon $T$, number of users $N$, and number of arms $K$.
Set $T_{exp} = 10C^{2/3}K^{1/3}T^{2/3}(\log NKT)^{1/3}$.
For each user $i \in [N]$ and arm $j \in [K]$ initialize two variables $n_{ij} = 0$ and $\sigma_{ij} = 0$.
**for** $r \leftarrow 1$ **to** $\lceil T_{exp}/(2C + 2) \rceil$ **do**

> Divide $[N]$ arbitrarily into $S$ groups $G_1, \ldots, G_S$ of size $C + 1$ (adding all remaining users to the last group $G_S$).
> For each group $G_s$, set $j_s = (r \bmod K)$.
> Run the feedback-eliciting sub-algorithm on the assignment $\pi$ induced by the groups $G_s$ (where if $i \in G_s$ then $\pi(i) = j_s$).
> **for** $i \leftarrow 1$ **to** $N$ **do**
>> $n_{i,\pi(i)} \leftarrow n_{i,\pi(i)} + 1$
>> $\sigma_{i,\pi(i)} \leftarrow \sigma_{i,\pi(i)} + \hat{\mu}_{i,\pi(i)}$. Here $\hat{\mu}_{i,\pi(i)}$ is the unbiased estimate for $\mu_{i,\pi(i)}$ produced by the feedback-eliciting sub-algorithm.
>
> **end**

**end**
**for** *remaining rounds $t$* **do**

> Match user $i$ to $\arg\max_j \sigma_{ij}/n_{ij}$.

**end**

---

Algorithm 2 follows the standard pattern of Explore-Then-Commit algorithms (see e.g. Chapter 6 of Lattimore and Szepesvári [2020]). For approximately $O(T^{2/3})$ rounds, we run the feedback-eliciting sub-algorithm on random assignments from users to arms (albeit ones which are chosen to guarantee each arm with any users matched to it has at least $C + 1$ users matched to it), getting unbiased estimates of the means $\mu_{i,j}$. For the remaining arms, we match each user to their historically best-performing arm.

**Theorem 2.** *Algorithm 2 incurs an expected regret of at most $\tilde{O}(NC^{2/3}K^{1/3}T^{2/3})$ for the anonymous bandits problem.*

### 3.6   Lower bounds

We finally turn our attention to lower bounds. In all of our lower bounds, we exhibit a family of hard distributions over anonymous bandits problem instances, where any algorithm facing a problem instance randomly sampled from this distribution incurs at least the regret lower bound in question.

We begin by showing an $\Omega(N\sqrt{CKT})$ lower bound that holds even in the presence of a user-cluster assumption (and in fact, even when $U \geq K(C+1)$). Since Algorithm 1 incurs at most $\tilde{O}(NC\sqrt{KT})$ regret for $U \geq K(C + 1)$, this shows the regret bound of our algorithm is tight in this regime up to a factor of $\sqrt{C}$ (and additional polylogarithmic factors).

**Theorem 3.** *Every learning algorithm for the anonymous bandits problem must incur expected regret at least $\Omega(N\sqrt{CKT})$, even when restricted to instances satisfying $U \geq K(C + 1)$.*

We now shift our attention to settings where there is no guaranteed user-cluster assumption. We first show that the $O(T^{2/3})$ dependency of the algorithm in Section 3.5 is necessary.

**Theorem 4.** *There exists a family of problem instances satisfying $N = C + 1$ where any learning algorithm must incur regret at least $\Omega(T^{2/3})$.*

Finally, we show the assumption in Section 3.5 that $N > C$ is in fact necessary; if $N = C$ then it is possible that no algorithm obtains sublinear regret.

**Theorem 5.** *There exists a family of problem instances satisfying $N = C$ where any learning algorithm must incur regret at least $\Omega(T)$.*

## 4  Simulations

Finally, we perform simulations of our anonymous bandits algorithms – the explore-then-commit algorithm (Algorithm 2) and several variants of Algorithm 1 with different decomposition algorithms – on synthetic data. We observe that both the randomized decomposition and LP decomposition based variants of Algorithm 1 significantly outperform the explore-then-commit algorithm and the greedy decomposition variant, as predicted by our theoretical bounds. We discuss these in more detail in Section B of the Supplemental Material.

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
