# OpenReview forum: "Anonymous Bandits for Multi-User Systems"
_NeurIPS.cc/2022/Conference — NeurIPS 2022 Accept_

### Official Review · Reviewer_ex8q · 2022-07-10

**Rating:** 7
**Confidence:** 2
**Soundness:** 3 good
**Presentation:** 2 fair
**Contribution:** 3 good

**Summary:**

Authors consider the problem of online learning in multi-user systems under a version of anonymity inspired by $k$-anonymity. In their model, a centralized learner assigns $N$ users to $K$ arms using groups of size at least $C$ (assigning all users in a group to the same arm) and can only observe this group’s aggregate rewards. The goal is to maximize total rewards while at the same time preserving the anonymity of individual rewards from users.

Their algorithms depend on a sub-algorithm for matching from users to arms and recovering an unbiased estimate of each user’s reward. The key idea is as follows: By getting aggregate rewards from a group (assigned to an arm) including a target user and by getting aggregate rewards from the group excluding the user, they can estimate the mean reward of the user to the arm.
Also, their algorithm needs the anonymous decompositions of bipartite graphs in which each entry represents the number of assignments from a user to an arm. The graph needs to be constructed to satisfy that the number of users assigned to an arm is larger than or equal to $C+1$.

They propose an algorithm using BaSE algorithm and Decompose for constructing anonymous decomposition. This algorithm achieves $\tilde{O}(NC\sqrt{\alpha KT})$ under the user-cluster assumption $U$. Also, they provide several algorithms for constructing anonymous decompositions. Lastly for the case of the absence of user-clustering assumption, using the explore-then-commit approach with the feedback-eliciting sub-algorithm, they achieve $\tilde{O}(NC^{2/3}K^{1/3}T^{2/3})$ regret bound.  They provide a lower bound of $\Omega(N\sqrt{CKT})$ under the user cluster assumption, and $\Omega(T^{2/3})$, otherwise.


**Questions:**

- I was wondering whether the $C$-anonymity could preserve the privacy of users. According to the suggested algorithms, they can estimate each mean reward which implies that the algorithms can estimate user preference.

- I was wondering whether the suggested algorithm with the anonymity polytope method can achieve Theorem 1.

- Could you explain how the approximation ratio $\alpha(C,U)$ can be measured?


**Limitations:**

- It would be better to discuss the gap between the regret upper and lower bound such as where does this gap come from?
- It would be better to explain the approximation ration $\alpha(C,U)$ in more detail such as how can we measure this?


**Strengths And Weaknesses:**

**Strengths**
- The problem setting with k-anonymity seems to be novel in bandit research

- They provide regret lower bounds for this problem and propose algorithms with regret analysis by achieving tight regret bounds.

**Weakness**
- I have a concern about whether the problem setting is suitable or not for preserving the privacy of users which is the main motivation of this paper. According to their algorithms, they utilize feedback-eliciting sub-algorithms to estimate mean rewards for user-arm pairs. This implies that the algorithm can recover the mean reward (preference) of each user so that privacy may not be preserved. On the other hand, for example, in the differential private linear contextual bandits [1], bandit algorithms were proposed by adding noise to context and reward information such that the server cannot estimate the mean rewards.

[1] Shariff, Roshan, and Or Sheffet. "Differentially private contextual linear bandits." Advances in Neural Information Processing Systems 31 (2018).


- Another concern is that it may not be clear that there exists a method of constructing groups for achieving Theorem1. Even though the authors discuss some methods such as a greedy method and the anonymity polytope, it would be better to discuss clearly whether the anonymity polytope can achieve the result of theorem 1.

---

> ### Author Response · Authors · 2022-08-02
> **Response to Reviewer ex8q**
>
> Thank you for your careful and thoughtful review of our paper. In response to your questions:
>
> Re: preserving the privacy of users. Note that the notion of anonymity we study is an event-level notion of privacy, intended to protect individual user decisions (e.g. whether or not the user clicked on the ad assigned to them on round t) and not general information (what types of ads user i is interested in). In fact, it is easy to observe that it is impossible to provide a private multi-armed bandits mechanism (under differential privacy or C-anonymity) with sublinear regret without revealing the optimum arm for the user. A simple example is where the value of one arm is 1 and the value of all other arms are 0. In this case, any algorithm with regret better than T/2 pulls the optimum arm more than half of the time and reveals it.
>
> See also our response to Q2 of Reviewer 1FZP and lines 29-40 of our paper (i.e., “the fact that the website assigns a user a specific type of ad is not something we intend to hide from the website. However, the decision to click on the ad is made by the user. We intend to protect these individual decisions, while allowing the website to learn what general types of ads each user is interested in.”)
>
> Re: achieving Theorem 1 via the anonymity polytope. Yes, this is precisely the goal of Section 3.4; it gives examples of anonymous decomposition algorithms that are used as a blackbox by the algorithm of Theorem 1.
>
> Re: measuring the approximation ratio alpha(C, U). Different anonymous decomposition algorithms give different approximation ratios alpha(C, U). Decomposition based on the anonymity polytope gives the ratios presented in lines 270 and 271.

---

> > ### Comment · Reviewer_ex8q · 2022-08-04
> > **Response to authors**
> >
> > Thank you for your response to my comments. All my concerns are resolved especially about the motivation of preserving privacy. Therefore, I raised my rating.
> >
> > For me, it took some time to understand how the algorithm works because the concept of batches, assignments, and rounds was confusing, and the algorithm contains many sub-algorithms such as Decompose, feedback-soliciting sub-algorithm, and BaSE.
> >
> > I think it would be better if there is a way to explain the individual concepts and the integrated view of the algorithms more clearly.

---

### Official Review · Reviewer_TxQ1 · 2022-07-12

**Rating:** 7
**Confidence:** 3
**Soundness:** 3 good
**Presentation:** 3 good
**Contribution:** 3 good

**Summary:**

This paper studies a new framework for multi-user multi-armed bandits that provides user anonymity. At each round, a centralized learner assigns $N$ users to $K$ arms and the reward for each user-arm pair is drawn from some fixed, unknown distribution. However, the learner can obtain information about the rewards and use it for learning only if the assignment conforms to the notion of $C$-anonymity. This notion requires partitioning the users into groups of size $C$, assigning all users in a particular group to the same arm and only observing the group's aggregate rewards for the assigned arm.

The goal of this work is to use the aggregated rewards received per arm in each round to minimize the group cumulative regret. In order to do so, the authors proposed regret minimization algorithms in two different regimes: one under the assumption for each arm $j \in [K]$, there are at least $C+1$ users for which arm $j$ is optimal and the other without this assumption. They also evaluate the tightness of the regret guarantees of the proposed algorithms by providing lower bounds for both regimes.

This paper uses ideas from batched bandits in Gao et. al. [2019] to reduce the online learning problem at hand to a combinatorial optimization problem involving decomposition of a bipartite weighted graph into a set of degree-constrained bipartite graphs.



**Questions:**

Please address the concerns in the weaknesses of the paper in as much detail as possible.

Minor comments:
1. Inconsistent notation: please replace $n$ with $N$ for the number of users (for example, in pseudocode of Algorithm 1)
2. In line 215, shouldn't $j^*(i) = \arg \max_{j} \mu_{i, j}$?
3. In line 222, it should read as follows: a user-cluster assumption of $U$ directly implies that $|B_j| \geq U$ "with high probability" (missing in the current version).
4. In the numerical results in the appendix, please run the algorithms for at least 30 independent runs for obtaining reasonably tight confidence bounds.

**Limitations:**

Yes

**Strengths And Weaknesses:**

Strengths:
1. The problem setup is well-motivated in the introduction with the example of displaying online advertisements to users.
2. This paper is well-written and easy to follow.
3. Related work is covered in detail.
4. In the process of obtaining the first non-trivial sub-linear regret guarantees for the anonymous MAB problem, the authors have creatively used the ideas from combinatorial optimization and convex geometry to come up with efficient approximation algorithms for solving the problem of decomposing a bipartite weighted graph into a set of degree-constrained bipartite graphs (which could be of independent interest in the optimization and algorithms community).

Weaknesses:
1. A potential downside of the applicability of this setting would be in cases where users are not active at all times, and as a result, the learner may not conform to the notion of $C$-anonymity for obtaining feedback about the arms.
2. As claimed in the paper, the user would like feedback on each of the arms (for example, in lines 194-196). This doesn't seem correct, because the user seems to be a source of data (rather than an agent) in this setting, and it is the learner which requires the feedback about the arms.
3. The result in Lemma $1$ seems incorrect: shouldn't $D_b . \mathbb{E}[\max_{j \in A_b} \Delta_j] = \widetilde{O}(\sqrt{K D_b})$ instead of $\widetilde{O}(\sqrt{K T})$?

---

> ### Author Response · Authors · 2022-08-02
> **Response to Reviewer TxQ1**
>
> Thank you for your careful and thoughtful review of our paper. In response to the concerns in the weaknesses:
>
> W1: It is true that in our model each user is assumed to be active in each time step. It is an interesting future direction to extend our results to the case where users may be inactive in some rounds; we believe that the work in this paper will still be relevant in this expanded setting (for example, we believe that it should be relatively straightforward to extend our algorithms to the case where each user is active with some fixed probability, at the cost of some loss in the parameter U).
>
> W2: Thanks for pointing this out; our wording is suboptimal in these places and we will change it (e.g., lines 194-196 should read “the number of times that we would like feedback for that user on each of the m arms”).
>
> W3: We think Lemma 1 is correct as written. Note that your bound is stronger (sqrt(KD_b) <= sqrt(KT)) and may be true, but we only ever use the fact that this quantity is O~(sqrt(KT)).
>
> Minor comments: Thanks for pointing these out. We will correct them (and regenerate plots with >= 30 runs).

---

> > ### Comment · Reviewer_TxQ1 · 2022-08-04
> > **Response to authors**
> >
> > I thank the authors for responding to my comments. I am satisfied with their responses and have decided to keep my score. Good luck!

---

### Official Review · Reviewer_1FZP · 2022-07-12

**Rating:** 7
**Confidence:** 3
**Soundness:** 4 excellent
**Presentation:** 3 good
**Contribution:** 3 good

**Summary:**

In this paper, the author(s) address a variant of the multi-arms bandit problem. In this setting, there is the need to keep the user's choices anonymous to the learner. In this regard, the author(s) design a feedback-eliciting sub-algorithm that will allow the learner to obtain the user's rewards without revealing the user's choice. Then the authors present an algorithm where each user will run BaSE (Gap et al 2019) independently.


**Questions:**

Q1: It would be useful for completeness if you spelled out that $K\geq N$. I'm assuming that how big $C, N, K$ are in comparison to each other does not really play an important role in the analysis, does it?

Q2: In your Feedback-eliciting sub-algorithm, one gets a sample of the total reward of a group containing the user, and a sample of the total reward of the same group without that on user. Is How does one know that this is a anonymous procedure; namely, is there no way to reverse-engineer the process and discover the user's choice? I think the steps 1 to 5 and Lemma 2 address this, but could you make it more explicit?

Q3: Could you please make sure there is a consistency in how you list the references and adding hyperlinks to the file.

Q4: Could you provide a simple example of a U-batched graph? I think this would help with the presentation

Q5: You tried different anonymous decomposition algorithms. Is there one that would be more "better" than other? or is it irrelevant?

Minor suggestions

Line 67: Notation $\tilde{O}$ is used before it is defined.

Line 173: development/analysis --> development and analysis

**Limitations:**

I did not see any specific discussion of limitations or potential negative societal impacts. If I'm mistaken, could you tell me where these are being addressed?



**Strengths And Weaknesses:**

Strengths
The paper has an interesting setting that is readily applicable to industry. For example, in A/B testing, this set up would protect the anonymity of the choice of the user. In today's world, there is the desire to preserve a certain level of anonymity, which makes this paper's setup both interesting and relevant.

The authors accomplish several things in the paper: They talk about how to partition the set of users in a way that their choices would remain anonymous. I thought their schemes to produce anonymous decompositions was interesting in their own right as a combinatorial problem. The authors also provide theoretical guarantees backing up the efficiency of their low-regret algorithm for anonymous bandits.

Weakness
I think this is overall a good paper. I have one minor comment relating to the formatting. There seems to be an inconsistency about how references are cited. Sometimes the author(s) use the format "[Author(s), year]" and sometimes "Author [Year]." The author(s) might want to consider going through the draft to make sure the references are consistent. There also doe snot seem to be a hyperlink from the references in text pointing to said reference in the bibliography.

---

> ### Author Response · Authors · 2022-08-02
> **Response to Reviewer 1FZP**
>
> Thank you for your careful and thoughtful review of our paper. In response to your specific questions:
>
> Q1: We think you probably mean N (number of users) >= K (number of arms). This is true whenever we have a user-cluster assumption U, which immediately implies N >= KU; we will spell out this inequality explicitly in the paper. In settings without user-clustering (Section 3.5), the relative size of N and K does not matter (and in general relative sizes do not matter unless pointed out explicitly, e.g. the N >= C+1 requirement in Section 3.5).
>
> Q2: Our feedback-eliciting sub-algorithm does preserve our notion of anonymity (as defined in Section 2.1). Note that in the notion of anonymity we study (“event-level anonymity”), it is okay for us to learn which arm (e.g., which type/topic of ad) a user values most -- what must remain private is the specific rewards the user obtained in specific rounds (e.g., whether or not a user clicked on the ad shown to them at time t). If we observe the aggregate reward of a large group that contains a user in one round, and then in a later round observe the aggregate reward of a large group that does not contain this user, this preserves this notion of anonymity -- it is impossible to conclude that the user clicked on the ad in either case. By subtracting the two aggregate rewards, we only get a very noisy estimate about whether the user likes arms of this type (since rewards in different rounds are independent).
>
> Q3: We will fix the inconsistency in reference formatting, thanks for pointing this out!
>
> Q4: One very simple U-batched graph is a bipartite graph with N=K*U users on the left and K arms on the right, where each arm is matched to U distinct users (and one can make this more complicated by adding any additional subset of edges to this). We will add an example like this to the paper.
>
> Q5: This is discussed in more detail in Appendix B of the Supplementary Material, but a short summary is that both linear programming-based anonymous decomposition (based on the anonymity polytope techniques 3.4.2) and randomized anonymous decomposition (described at the beginning of 3.4.2) both do very well in practice (significantly better than the greedy anonymous decomposition algorithm of 3.4.1).
>
> Limitations: As we mention in the Checklist, we don’t foresee any negative societal impacts of this work (if anything, we feel the potential societal impacts are positive, as we provide new theoretical tools for achieving anonymity and privacy in learning systems).
>
> Minor suggestions: Thanks for pointing these out, we will correct them.

---

> > ### Comment · Reviewer_1FZP · 2022-08-09
> > **Thanks!**
> >
> > Thank you for your comments. I have read them and my rating stands. Good luck!

---

### Official Review · Reviewer_f33e · 2022-07-18

**Rating:** 6
**Confidence:** 3
**Soundness:** 3 good
**Presentation:** 3 good
**Contribution:** 3 good

**Summary:**

The authors study a new multi-armed bandit problem, where there are $N$ users and $K$ arms. At each time $t$, a central entity allocates  each user $i$ to an arm $j$ (several users may be allocated the same arm), and a stochastic 1-subgaussian reward with mean $\mu_{i,j}$ is obtained. Not all mappings between users and arms are allowed though: for all arms $j$ the number of users allocated to this arm must be greater than a fixed integer $C \ge 1$.This constraint is used to model a privacy requirement, and represents a relaxed notion of privacy with the standard notion of differential privacy.

The authors then propose several algorithmic techniques to solve the problem, and provide theoretical regret upper bounds. Let $U$ be the minimal number of users that share the same optimal arm, When $U \ge C+1$ their algorithm has a regret of $O( N C K \sqrt{T} )$ and when $U \ge K(C+1)$ their algorithm has a regret of $O( N C \sqrt{K T} )$.

**Questions:**

- Is it possible to compare between the proposed algorithms and linear bandit algorithms applied to this problem, in terms of computational complexity and regret guarantees ?
- If one were to use a modified definition of regret, would the proposed results significantly change ?



**Limitations:**

See above.

**Strengths And Weaknesses:**

Strenghts
- The paper is well written and easy to follow
- The authors give a thorough treatment of the problem with algorithms, numerical experiments, regret upper bounds and lower bounds
- The idea of exploring relaxed notions of privacy (with respect to differential privacy) seems very important. While differential privacy is theoretically well understood, in many practical settings it is too stringent, and studying other measures of privacy seems critical for practical applications.

Weaknesses
- **Casting the problem as a linear bandit** Upon close inspection, the problem studied by the authors is a linear bandit (as well as a combinatorial bandit too). To cast the problem as a linear bandit denote by
$$
X = \\{ x \in \\{0,1\\}^{N \times K} :  \sum_{i \in [N]} x_{i,j} \ge C \\;\\;\\; \forall j  \\} \subset \\{0,1\\}^{N \times K}
$$
the set of available decisions, where $x_{i,j} = 1$ if user $i$ is allocated to arm $j$ and $x_{i,j} = 0$ otherwise, and let the reward obtained by the learner when choosing decision $x$ be
$$
        \sum_{i \in [N]} \sum_{j \in [K]} x_{i,j} \mu_{i,j} +  \sum_{i \in [N]} \sum_{j \in [K]} x_{i,j} Z_{i,j}
$$
where $Z_{i,j}$ are $1$ sub-gaussian with mean $0$. The reward can be rewritten as
$$
      {1 \over \sqrt{N K}} \sum_{i \in [N]} \sum_{j \in [K]} x_{i,j} \mu_{i,j} + W
$$
where $W$ is 1-subgaussian with mean $0$. So indeed this is a standard linear bandit (a combinatorial bandit as well) in dimension $N \times K$, where the set of decisions is $X$, the unknown mean reward vector is ${1 \over \sqrt{NK}} \mu$ and the reward distribution is $1$ subgaussian. Now this implies that any algorithm for linear bandits can be applied to the problem at hand, for instance this implies that one can always obtain regret scaling as $O( N K \sqrt{T})$, and this should work irrespective of the value of $C$ and $U$, while those numbers play a critical role in the regret upper bounds at hand.

- **Definition of regret** The authors define the regret as
$$
T \sum_{i \in [N]} \max_{j \in [K]} \mu_{i,j} - \sum_{t \in [T]} \sum_{i \in [N]} E(  r_{i,t} )
$$
This is the difference between the reward obtained by an oracle that allocates the optimal arm to each user and the learner. Now I feel that this definition is problematic, in the sense that the oracle is allowed to take actions that cannot be taken by the learner. And indeed, in some regimes, the regret cannot be made sub-linear. It seems to me that it would make more sense to define the regret as (for $X$ see definition above):
$$
T \max_{x \in X} \left( \sum_{i \in [N]} \sum_{j \in [K]} x_{i,j} \mu_{i,j} \right) - \sum_{t \in [T]} \sum_{i \in [N]} E( r_{i,t})
$$
where the oracle is only allowed actions that are also  allowed to the learner.  With this definition the regret can always be made sublinear, and indeed, there exists algorithms with sublinear regrets (since the problem is a linear bandit).

---

> ### Author Response · Authors · 2022-08-02
> **Response to Reviewer f33e**
>
> Thank you for your careful and thoughtful review of our paper.
>
> Re: casting the problem as a linear bandit -- there are some subtle differences between the problem we solve in the paper and the problem you describe in the summary, which make it tricky to model and solve our problem via linear / combinatorial bandits. The model you describe solves the problem where we must assign at least C users to every arm. However, in our problem, it is permitted to assign 0 users to some arms (this does not violate any sense of privacy). In fact, we actually allow for an arbitrary assignment of users to arms -- the constraint of assigning at least C user to arms only affects our *feedback* (for arms where we assign fewer than C users, we are not allowed to measure the average reward of users on that arm). This expands the polytope you mention to what we call the C-anonymity polytope in Section 3.4.2, which appears hard to optimize over (and hence apply linear / combinatorial bandits methods). Nonetheless, thanks for bringing these points up; we will add a discussion of linear / combinatorial bandit methods (and this related subtlety) to the paper.
>
> Similarly, in the definition of regret, our model actually *does* allow the learner to assign the optimal arm to each user: learners are allowed to assign users to arms arbitrarily, and the privacy constraint only affects the feedback they receive. Our linear regret lower bounds come from the fact that in some settings, it is impossible for a learner to learn this optimal assignment. It is an interesting question to understand the gap between our algorithms and the “best private learner” (although this is a bit ambiguous), but we think the notion of regret we study is a bit more natural and interesting in this setting (for instance, it captures the “cost of anonymity”; one way of interpreting our results is that the cost of being anonymous is sublinear in a variety of settings).

---

### Meta-Review · Area_Chair_6ZpL · 2022-08-26

**Recommendation:** Accept
**Confidence:** Certain

**Metareview:**

The paper introduces a conceptually new variation of the standard multi armed bandit problem motivated by anonymity and ensuring privacy of user responses. This is likely to open up a new avenue for research into formulations of sequential decision making involving anonymity constraints.

The reviewers' opinions were insightful, including a pertinent comment on the connection to casting the framework as a linear (combinatorial) bandit. The reviews were responded to in detail, and finally all the reviewers are unanimous in their positive view of the paper. Thus the paper deserves to be accepted. I request the author(s) to please use the additional page in the final version to discuss the subtleties of the comparison with linear bandits, in order to make this paper more complete in its statement.

**Award:**

No

---

### Decision · Program_Chairs · 2022-09-14

Accept